# Noncoding RNAs of Extracellular Vesicles in Tumor Angiogenesis: From Biological Functions to Clinical Significance

**DOI:** 10.3390/cells11060947

**Published:** 2022-03-10

**Authors:** Miao Hu, Juan Li, Chen-Guang Liu, Robby Miguel W. J. Goh, Fenggang Yu, Zhaowu Ma, Lingzhi Wang

**Affiliations:** 1School of Basic Medicine, Health Science Center, Yangtze University, Jingzhou 434023, China; 201706692@yangtzeu.edu.cn (M.H.); 201972551@yangtzeu.edu.cn (C.-G.L.); 2Department of Epidemiology, School of Public Health, Cheeloo College of Medicine, Shandong University, Jinan 250012, China; lijuan107@hust.edu.cn; 3Watford General Hospital, Watford Herts WD18 0HB, UK; r.goh@nhs.net; 4Institute of Life Science, Yinfeng Biological Group, Jinan 250000, China; yufenggang@yinfeng.com.cn; 5Department of Pharmacology, Yong Loo Lin School of Medicine, National University of Singapore, Singapore 117600, Singapore; 6Cancer Science Institute of Singapore, National University of Singapore, Singapore 117599, Singapore; 7NUS Centre for Cancer Research, Yong Loo Lin School of Medicine, National University of Singapore, Singapore 117599, Singapore

**Keywords:** extracellular vesicles, noncoding RNAs, tumor angiogenesis, cancer biomarkers, drug resistance

## Abstract

Extracellular vesicles (EVs) act as multifunctional regulators of intercellular communication and are involved in diverse tumor phenotypes, including tumor angiogenesis, which is a highly regulated multi-step process for the formation of new blood vessels that contribute to tumor proliferation. EVs induce malignant transformation of distinct cells by transferring DNAs, proteins, lipids, and RNAs, including noncoding RNAs (ncRNAs). However, the functional relevance of EV-derived ncRNAs in tumor angiogenesis remains to be elucidated. In this review, we summarized current research progress on the biological functions and underlying mechanisms of EV-derived ncRNAs in tumor angiogenesis in various cancers. In addition, we comprehensively discussed the potential applications of EV-derived ncRNAs as cancer biomarkers and novel therapeutic targets to tailor anti-angiogenic therapy.

## 1. Introduction

Under physiological and pathological conditions, almost all types of cells can secrete membrane vesicles collectively referred to as small extracellular vesicles (sEVs) and large EVs (lEVs) [1,2]. Exosomes, small EVs, play a crucial role in intercellular communication by transporting proteins, mRNAs, and ncRNAs [2]. In the past decade, ncRNAs, including microRNAs (miRNAs), PIWI-interacting RNAs (piRNAs), long noncoding RNAs (lncRNAs), and circular RNAs (circRNAs), have been intensively studied in many human diseases, particularly cancers [3,4,5,6]. Studies have indicated that exosomal ncRNAs acting as versatile regulators at the transcriptional and post-transcriptional levels may contribute to cancer progression and metastasis [7,8]. Recently, it was found that EV-derived ncRNAs exhibit a diverse range of biological functions in cancer hallmarks, including tumor angiogenesis.

Angiogenesis, a multi-step process involving the formation of new blood vessels from existing vasculature, occurs during the natural growth, development, and progression of diseases, such as solid cancer [9]. Tumor angiogenesis participates in the whole process of tumorigenesis, proliferation, and metastasis, from the initial stages of pre-tumor to carcinoma in situ and, subsequently, to malignant tumor [10]. Proliferation and metastasis of tumor cells both depend on the adequate supply of oxygen and nutrients and the elimination of waste products [11]. Newly formed blood vessels can fulfil this requirement. Angiogenesis is caused by the imbalance between pro- and anti-angiogenic factors resulting from the excessive production of vascular endothelial growth factor (VEGF) induced by hypoxia [12]. Studies have reported that EV-derived ncRNAs regulate gene expression via a diverse range of regulatory mechanisms, contributing to diverse cancer phenotypes and tumor angiogenesis.

Recently, certain reviews have focused on the general role of EVs or miRNAs in angiogenesis [13,14,15]; however, the functions and mechanisms of EV-derived ncRNAs have not been comprehensively explored in tumor angiogenesis. In this review, we summarized the functions of EV-derived ncRNAs in tumor angiogenesis and highlighted their potential clinical applications in cancer diagnosis and therapy.

## 2. Biology of EVs, Angiogenesis, and ncRNAs

### 2.1. Characteristics of EVs and ncRNAs

EVs are extracellular structures enclosed in a lipid bilayer that can be secreted by almost all known cell types [16]. Several studies have indicated that EVs play an essential role in intercellular communication between tumor cells and the tumor microenvironment (TME). EVs are highly heterogeneous; therefore, their characterization and classification are crucial for further research to avoid generating inconclusive results. Based on their size, biogenesis, and release pathways, EVs can be broadly divided into three main subtypes: exosomes, microvesicles (MVs), and apoptotic bodies. Exosomes, also called small EVs, possess a diameter ranging from 30 to 150 nm and are derived from multivesicular endosomal pathways, which are formed by inward budding of the endosomal membrane in a process that sequesters particular proteins and lipids [17]. On the contrary, MVs are generated by regulated outward budding of the plasma membrane [18]. The mechanisms of exosomal biogenesis involve multiple factors, and the most well-known regulator is endosomal sorting complex required for transport (ESCRT) [18]. Exosomal biogenesis involves inward budding of the plasma membrane to form endosomes, leading to production of multivesicular bodies (MVBs), fusion of MVBs with the plasma membrane, and release of exosomes into the extracellular space. A core component of this mechanism is the ESCRT machinery, which consists of four protein complexes and auxiliary proteins that bind to future exosome cargoes and form intraluminal vesicles that incorporate those cargoes [19,20]. Several studies have found that exosomes can be formed despite the depletion of the ESCRT complex, which reveals an ESCRT-independent approach [21]. ESCRT-independent exosomal biogenesis is regulated by sphingolipid ceramide, which is produced from the hydrolysis of sphingomyelin by neutral sphingomyelinase 2 [17]. The contents of EVs include various nucleic acids, lipids, and proteins [22]. ncRNAs carried by EVs can regulate various physiological and pathological processes through multiple mechanisms (Figure 1).

In the past few decades, the development of high-throughput sequencing technology has indicated that the transcription rate of the human genome generally exceeds 70%. However, <2% of the transcript is translated into proteins; most human transcriptomes are ncRNAs. Emerging studies have shown that despite being ‘transcriptional junk’, ncRNAs, such as miRNAs, piRNAs, circRNAs, snoRNAs, and the attractive lncRNAs, play a versatile role in manipulating gene expression [23]. miRNAs, a type of endogenous small RNA with a length of 20–24 nucleotides (nt), have many essential adjustable functions in cells. By complementarily binding to the 3’-untranslated region (UTR) of targeted mRNAs, miRNAs act as regulators of gene expression, thereby inhibiting post-transcriptional gene expression [24,25,26]. Recent studies have reported that miRNAs are selectively sorted into EVs and participate in intercellular communication in the TME [27]. In addition, EV-derived miRNAs in biofluids can be used as ideal biomarkers for various types of tumors due to their easy accessibility, high abundance, and good stability [28]. piRNAs, a type of small RNA with a length of 21–35 nt, specifically interact with PIWI protein to perform multifaceted functions in germline development and somatic tissues [29,30,31].

In addition to small RNAs, large ncRNAs also participate in gene regulation in various biological processes. lncRNAs, collectively referred to as transcripts with more than 200 nt, have limited potential to encode proteins [32]. They perform their functions through multiple molecular and cellular mechanisms, such as interacting with epigenetic factors or TFs to modulate gene transcription, sequestering miRNAs, interacting with proteins, and encoding functional small peptides [33,34,35,36]. In addition, lncRNAs can also be selectively sorted into EVs and participate in cell-to-cell communication in the TME [37]. circRNAs, generated by a particular form of alternative splicing called back-splicing, regulate gene transcription and translation by interacting with DNAs, RNAs, and proteins [38]. Emerging studies have indicated that circRNAs participate in multiple physiological and pathological processes, including tumor angiogenesis [39,40,41].

### 2.2. EV-Derived ncRNAs: New Players in Tumor Angiogenesis

Angiogenesis is a multi-step process and has two types: sprouting and intussusceptive angiogenesis [42]. Various types of cells, including endothelial cells (ECs), tumor cells, stromal cells, and immune cells, regulate angiogenesis in the blood vessels. In addition, some regulatory and signalling molecules govern angiogenesis, including growth factors (e.g., VEGF, PDGF, FGF, and EGF) and transcription factors, such as HIFs [43,44,45]. Because angiogenesis is crucial for tumor growth and metastasis, targeting tumor-associated angiogenesis is a promising strategy for cancer treatment [46,47,48]. Currently, anti-angiogenic therapies targeting VEGF and VEGFR are used for the treatment of various tumors [49].

Several studies have indicated that EVs can be used as ncRNA carriers to play diverse roles in regulating tumor hallmarks, including angiogenesis. For example, in non-small cell lung cancer (NSCLC), RCAN1.4 has been identified as a target of miR-619-5p, and its suppression promotes angiogenesis [50]. The exosomal lncRNA FAM225A upregulates NETO2 and FOXP1 expression by sponging miR-206 to accelerate oesophageal squamous cell carcinoma (ESCC) progression and angiogenesis [51]. In addition, circSHKBP1 sponges miR-582-3p to enhance HUR expression and VEGF mRNA stability, which promotes angiogenesis and gastric tumor progression [52]. Moreover, emerging studies have indicated that EV-derived ncRNAs can regulate tumor angiogenesis by influencing a wide variety of tumor-associated molecules. The functions and mechanisms of EV-derived ncRNAs in tumor angiogenesis are summarized in Table 1. These studies suggest that EV-derived ncRNAs play an essential role in tumor angiogenesis. However, new technologies and animal models are required to further investigate the precise mechanisms of EV-derived ncRNAs in the regulation of tumor angiogenesis.

## 3. Emerging Functions of EV-Derived ncRNAs in Tumor Angiogenesis

The TME is composed of the extracellular matrix, blood vessels, immune cells, stromal cells, and bioactive substances, all of which enrich the local environment of tumor cells. It plays an integral role in tumor migration, tube formation, cell adhesion, invasion, proliferation, angiogenesis, and treatment resistance [104]. EVs contain various signalling molecules that are important for the reprogramming function of intercellular communication in the TME [19,105]. Tumor-derived EV-derived ncRNAs can regulate the behaviour of surrounding immune and stromal cells, ultimately creating a suitable microenvironment for tumor growth [106]. In addition, immune and stromal cells secrete EV-derived ncRNAs in the TME, which, in turn, affects tumor progression [107]. In the subsequent sections, we have summarized the emerging roles of EV-derived ncRNAs in tumor angiogenesis and highlighted the crosstalk between tumor cells and the surrounding cells within the TME (Figure 2).

### 3.1. Tumor-Derived EVs Affect Tumor Angiogenesis by Transferring ncRNAs

The TME is a ‘complex society’ of many cell types and their extracellular matrices. Different immune molecules exist in the TME, which include macrophages, neutrophils, mast cells, dendritic cells (DCs), natural killer (NK) cells, and T and B lymphocytes [108]. In addition, the TME also consists of tumor stromal cells, including mesenchymal stromal cells (MSCs), fibroblasts, ECs, and pericytes [109,110]. Primary tumors recruit immune and stromal cells to their microenvironments to induce angiogenesis mediated by EV-derived ncRNAs, leading to increased tumor proliferation and metastatic potential (Figure 2).

EV-derived ncRNAs derived from tumor cells promote the M2 polarization of macrophages in the TME, thereby affecting angiogenesis. A study found that EVs derived from hypoxic lung cancer (LC) cells increased the polarization of M2 macrophages through the miR-103a/PTEN axis. The M2-type cytokines (IL-10) and cancer-promoting cytokines (VEGF-A) produced by M2-polarized macrophages further promoted cancer progression and angiogenesis [65] (Figure 2A). Similarly, the EMT transcription factor Snail directly stimulates the production of miR-21-rich tumor-derived exosomes (TEX). Furthermore, CD14^+^ human monocytes phagocytize exosomes containing miR-21, thereby inhibiting the expression of M1 markers, increasing the expression of M2 markers, and eventually promoting angiogenesis [111].

Cancer-associated fibroblasts (CAFs) differentiated from fibroblasts and MSCs were shown to promote tumor growth [112]. Exosome-mediated transfer of ncRNAs can also enhance the activation of CAF-promoting angiogenic transformation. The transmission of EV-derived ncRNAs from hepatocellular carcinoma (HCC) melanoma cells induced the transformation of fibroblasts to CAFs, leading to cancer-supporting angiogenesis. Zhou et al. demonstrated that cancer-derived exosomal miR-21 could convert hepatic stellate cells (HSCs) to CAFs by downregulating PTEN, thereby activating CAFs and promoting cancer angiogenesis by upregulating the expression of VEGF-α, MMP2, MMP9, bFGF, and transforming growth factor-β (TGF-β). [80]. In addition, exosomal miR-155 secreted by melanoma cells could activate the JAK2/STAT3 signalling pathway by reducing SOCS1 expression, thereby triggering the reprogramming of normal fibroblasts into pro-angiogenic CAFs. CAFs further increase the expression and secretion of VEGF-α, FGF2, and MMP9, thus promoting the proliferation, migration, and tube formation of ECs and angiogenesis in melanoma [113] (Figure 2A). Therefore, current evidence indicates that EV-derived ncRNAs influence tumor angiogenesis through diverse cell types, including macrophages and CAFs.

ECs receive various information from their environment, eventually leading to their rapid growth to form new blood vessels [114]. Accumulating evidence indicates that tumor-cell-derived ncRNAs may be packaged into EVs and transferred to recipient cells, resulting in the regulation of angiogenesis by influencing the proliferation and migration of ECs. In gastric cancer (GC), exosomes can function as vehicles to deliver different miRNAs (e.g., miR-155, miR-130a, and miR-135b), eventually enhancing the growth of blood vessels and promoting the metastasis of GC [53,54,55,56]. In colorectal cancer (CRC), exosomal miR-25-3p targets KLF2 and KLF4, thereby modulating the expression of VEGFR2, ZO-1 and Claudin-5 in ECs and promoting vascular permeability and angiogenesis [61]. In ESCC, miR-181b-5p is transferred from ESCC cells to vascular ECs through exosomes, as evidenced by angiogenesis induced by targeting PTEN and PHLPP2. Moreover, higher expression of miR-181b-5p in patients with ESCC predicts poorer overall survival [81]. In leukemia, exosomal miR-210 downregulates EFNA3, thereby enhancing tube formation in human umbilical vein endothelial cells (HUVECs) [72]. In multiple myeloma, the number of exosomes carrying miR-135b is significantly increased, and these exosomes directly inhibit factor-inhibiting hypoxia-inducible factor 1 (FIH-1) in ECs, resulting in the formation of endothelial tubes through the HIF–FIH signalling pathway [57]. In glioma, exosomes carrying miR-9 absorbed by vascular ECs can increase angiogenesis by directly targeting COL18A1, THBS2, PTCH1, and PHD3. This process can be activated by MYC and OCT4 [70]. In addition, another type of small ncRNA, piRNA, was also reported to regulate angiogenesis. piRNA-823 delivered by MM-derived EVs promoted angiogenesis by increasing the secretion of VEGF and IL-6 from ECs [103].

In addition to the small ncRNAs mentioned above, EV-lncRNAs and circRNAs derived from tumor cells also mediate tumor angiogenesis. In NSCLC, exosomal lncRNA-p21 modulates miRNA-containing exosomal cargoes and influences EC behaviour, thereby regulating angiogenesis and EC permeability [115] (Figure 2A). Exosomal lncRNAs (such as MALAT1 and RAMP2-AS1) are transferred to recipient HUVECs and eventually promote angiogenesis by increasing VEGF and VEGFR2 expression in epithelial ovarian cancer (OC) and chondrosarcoma, respectively [88,94]. In breast cancer (BC), exosomal SNHG1 promotes the proliferation, migration, and angiogenesis of HUVECs via the miR-216b-5p/JAK2 axis [92]. In pancreatic cancer (PC), exosomal UCA1 is delivered to HUVECs and promotes angiogenesis via the miR-96-5p/AMOTL2 axis [90]. In GC, exosomal X26nt decreases vascular endothelial cadherin (VE-cadherin) expression by directly binding to the 3’UTR of VE-cadherin mRNA, thereby increasing vascular permeability and angiogenesis in HUVECs [116]. Exosomal AC073352.1 secreted by BC cells is internalised by HUVECs and promotes angiogenesis by interacting with YBX1 and stabilizing its protein expression [93]. Exosomal H19 released by cancer stem cells promotes endothelial tube production by upregulating VEGF [97]. In addition, circRNA-100338 expression is high in both highly metastatic HCC cells and HCC cell-secreted exosomes. Exosomal circRNA-100338 enhances the pro-angiogenic ability and permeability of HCC cells by interacting with NOVA2, thereby affecting cancer proliferation (Figure 2A) [98].

Tumor-derived EV-derived ncRNAs can be useful indicators of tumor burden, prognosis, and treatment because they regulate many aspects of the angiogenic microenvironment. Therefore, specific EVs and their genetic cargoes in the TME may be used as biomarkers for the prevention and diagnosis of tumors or as novel targets for cancer treatment.

### 3.2. Immune and Stromal Cell-Derived EVs Affect Tumor Angiogenesis by Transferring ncRNAs

Tumors can release extracellular signals to create an angiogenic TME suitable for their growth and development. In turn, the diverse immune and stromal cells in the microenvironment can also manipulate tumor development and progression. EV-derived ncRNAs secreted by immune and stromal cells in the TME regulate tumor angiogenesis. For example, Zhang et al. found that miR-130b-3p enclosed in M2 macrophage-derived EVs was delivered to GC cells, thereby promoting angiogenesis via both the downregulation of MLL3 and upregulation of GRHL2 (Figure 2B) [84]. In another study, M2 macrophage-derived exosomes increased the blood vessel density of mouse pancreatic ducts in vitro and promoted tumor growth by targeting E2F2 [76]. In addition, miR-10a-5p derived from CAFs promotes angiogenesis and tumorigenesis in cervical squamous cell carcinoma by activating Hedgehog signalling by inhibiting TBX5 (Figure 2B) [83]. Pakravan et al. showed that the transfer of exosomal miR-100 from human bone marrow MSCs to BC cells downregulated the expression of VEGF in BC-derived cells by regulating the mTOR/HIF-1α signalling axis, thereby inhibiting angiogenesis (Figure 2B) [78]. miR-126 in MSC-derived EVs modulates the pro-angiogenic capacity of ECs by upregulating VEGF [66]. Therefore, fibroblasts, macrophages, and MSCs have been shown to exert synergistic effects to promote tumor angiogenesis in the TME.

EV-derived ncRNAs, as signalling molecules for communication between tumor cells and other components of the TME, play an important role in remodelling the TME to regulate tumor angiogenesis, thereby leading to tumor progression. However, the current research progress on EVs secreted by immune and stromal cells is far behind that of EVs secreted by tumor cells. Therefore, more studies are warranted to further understand the potential impact of EV-derived ncRNAs on the TME, particularly on tumor angiogenesis. To date, a majority of biological studies on EV-derived ncRNAs have been conducted using cell culture systems in vitro. However, it remains unclear whether EV-derived ncRNAs derived from artificial 2-dimensional cell culture systems can accurately reflect the biological conditions of the body. To address this concern, tumor organoids should be considered suitable models to investigate the potential impacts of EV-derived ncRNAs on the TME.

## 4. Signalling Pathways Regulated by EV-Derived ncRNAs in Tumor Angiogenesis

Angiogenesis requires the participation of various molecules and can be adjusted by multiple factors. Studies have revealed that a series of signalling molecules and pathways contribute to the angiogenic response [47]. To achieve a better understanding of regulatory pathways involved in angiogenesis, in this section, we have discussed the features and functional roles of EV-derived ncRNAs in angiogenesis-associated pathways, including the JAK/STAT, TGF-β, PI3K/AKT, and nuclear factor-κB (NF-κB) pathways.

The JAK/STAT and TGF-β pathways have been implicated in distinct essential cellular functions related to proliferation, survival, differentiation, and angiogenesis. STAT and SMAD are excessively activated in different tumor types and act as crucial signalling nodes in the TME [117,118]. A study revealed that the transfer of exosomal miR-210 derived from HCCs to ECs promoted tumor angiogenesis by targeting STAT6 and SMAD4 [73]. Abnormal TGF-β signal transduction may lead to various diseases, such as abnormal embryonic development, immune diseases, tissue fibrosis, and cancerous diseases. TGF-β regulates not only physiological angiogenesis but also angiogenesis in the later stages of tumor development [119]. A similar phenomenon also exists in hypoxic papillary carcinoma: hypoxic papillary thyroid carcinoma cells secrete exosomes rich in miR-21-5p, thereby promoting endothelial tube formation by inhibiting the expression of TGF-βI and COL4A1 [77]. In addition, Lawson J et al. found that miR-142-3p, a known cancer inhibitor in lung adenocarcinoma, was enriched in lung adenocarcinoma cell (LAC)-derived exosomes. The transfer of miR-142-3p from LACs to ECs accelerated angiogenesis by repressing TGFβR1 expression [64]. Another study revealed that exosomes promoted angiogenesis by transferring lnc-CCAT2 and lnc-POU3F3 to ECs, leading to the activation of VEGFA, TGF-β, bFGF, and bFGFR in glioma [87,120]. PTEN, a cancer suppressor gene, is a key mediator of the PI3K/AKT pathway and plays a vital role in the formation of normal blood vessels during tumor progression [121]. A recent study found that exosomal miR-205 secreted by OCs induced angiogenesis by silencing PTEN and subsequently activating its downstream AKT pathway. Moreover, PI3K or AKT inhibitors significantly delay cell migration and tube formation induced by exosomal miR-205 [68]. In glioma stem cells (GSCs), the expression of miR-26a increases, whereas that of PTEN decreases. Exosomal miR-26a facilitates the angiogenesis of human brain microvascular endothelial cells (HBMECs) by targeting PTEN to activate the PI3K/AKT signalling pathway [74]. Another study showed that nasopharyngeal carcinoma (NPC) cells secreted exosomes that enclose miR-9, which suppressed endothelial tube formation and migration by targeting MDK and modulating the PDK/AKT signalling pathway [69]. Emerging studies have indicated that EV-derived ncRNAs also orchestrate the crossroads of angiogenic signalling pathways. A recent study showed that exosomal miR-141-3p secreted by OCs promoted angiogenesis by upregulating the expression of VEGFR-2 in ECs, thereby activating the JAK/STAT3 and NF-κB signalling pathways [67].

To date, several studies have reported the underlying mechanisms of EV-derived ncRNAs in various signalling pathways and their regulatory functions in tumor angiogenesis to a certain extent. However, more studies are required to elucidate the precise molecular mechanisms of ncRNAs in regulating signalling pathways relevant to tumor angiogenesis. Furthermore, intervention in these signalling pathways may offer promising strategies for cancer treatment. More importantly, it is likely that other signalling pathways that induce tumor angiogenesis will be identified in the future. Understanding the mechanisms that mediate tumor angiogenesis will help scholars to further develop novel strategies and therapeutic approaches for inhibiting angiogenesis [122]. However, the crosstalk between angiogenesis-related signalling pathways remains unclear and warrants further investigation. Understanding these pathways may have significant implications in the development of novel anti-angiogenic cancer therapy.

## 5. Potential Clinical Applications of EV-Derived ncRNAs in Cancers

### 5.1. EV-Derived ncRNAs as Promising Tumor Biomarkers

EVs that possess great potential as disease biomarkers and therapeutic carriers have attracted increasing attention [123,124]. Biomarkers are molecules that can be used for diagnosis or prognosis. An ideal biomarker should have the following four most important characteristics: specificity, sensitivity, stability, and easy accessibility in a relatively non-invasive way. Most studies have focused on extracellular ncRNAs as potential biomarkers because they are stable and can be easily extracted from liquid biopsy, such as blood, urine, or other body fluids, using simple, sensitive, and relatively inexpensive assays. It has been reported that the quality of EV-derived ncRNAs is almost unaffected even after the samples are stored for many years because of the uniqueness of the source cell components and protection via encapsulation in the membrane [125,126]. Therefore, EVs are stable in circulation and under various storage conditions. We conducted a comprehensive literature search of EV-derived ncRNAs from different sources and found valuable biomarkers for the diagnosis and prognosis of multiple cancers (Figure 3A).

Researchers are interested in investigating candidate miRNAs, lncRNAs, and circRNAs carried by EVs that may serve as biomarkers for the diagnosis, prognosis, and treatment of tumors [7]. For example, in a study, ROC curve analysis demonstrated that plasma miR-601 and miR-760 could both be used as promising diagnostic biomarkers for advanced CRC [127]. In a study involving patients with CRC, Ogata et al. reported that the area under the curve (AUC) was 0.953, 0.948, and 0.798. In another study, exosomal miR-23a, miR-1246, and miR-21 could distinguish CRC (all stages) from the control area [128]. Furthermore, higher levels of plasma miR-320-EV and miR-126-EV in patients with high-risk LC can promote angiogenesis and can be significantly associated with poor overall survival [66]. A similar study showed that serum exosomes of patients with GC were rich in lncRNA ZFAS1. In addition, the upregulation of ZFAS1 was significantly associated with tumor lymphatic metastasis and TNM staging. These studies indicate that exosomal ZFAS1 may serve as a potential prognostic biomarker for GC [129]. Another study reported that exosomal ENSG00000258332.1 and LINC00635 could be used to differentiate patients with HCC from those with chronic hepatitis B with high specificity. Therefore, serum exosomal ENSG00000258332.1 and LINC00635, which are highly sensitive and can be obtained non-invasively, may be used as biomarkers for HCC [130]. Similarly, recent studies have reported serum exosomal circRNAs as novel and useful tools for the non-invasive diagnosis of cancer [131]. Exosomal circPRMT5 is highly expressed in the serum and urine of patients with bladder cancer and is closely related to tumor metastasis [132]. In addition, certain diagnostic clinical trials are currently underway. In one such trial, exosomal lncRNAs are isolated from serum samples for the diagnosis of lung cancer (NCT03830619).

A large number of studies have focused on the diagnostic, prognostic, and therapeutic significance of EV-derived ncRNAs in different tumor types. However, the specific role of EV-derived ncRNAs in angiogenesis-related diseases remains unclear. Furthermore, almost all studies have focused on cellular experiments and EV-ncRNA-associated applications in vitro; therefore, further studies are required to validate the findings for in vivo models. In addition, the potential of EV-derived ncRNAs as biomarkers remains to be further verified in multi-center, large-scale clinical trials.

### 5.2. EV-Derived ncRNAs as Potential Anti-Angiogenic Therapeutic Targets

Because of their negligible antigenicity, minimal cytotoxicity, and ability to bypass endocytic pathways and phagocytosis, EVs are considered ideal natural carriers for the delivery of ncRNAs [133]. In a study, engineered exosomes modified with DSPE–PEG2K–RGD loaded with miR-92b-3p produced synergistic anti-tumor and anti-angiogenesis effects with apatinib in nude mice models of abdominal tumors [3]. Another study showed that the delivery of miR-29a/c using cell-derived MVs inhibited angiogenesis in GC [134]. Furthermore, EV-derived ncRNAs have been demonstrated to be functional towards tumor hallmarks in different cell lines. Huang et al. showed that exosomes derived from HCC cells silenced with circRNA-100338 could significantly decrease the invasive ability of HCC cells. In addition, these exosomes could reduce cell proliferation, angiogenesis, permeability, vasculogenic mimicry (VM) formation ability of HUVECs, and tumor metastasis [98]. Bai et al. demonstrated that exosomal miR-135b secreted by GC cells inhibited the expression of FOXO1 protein and enhanced the growth of blood vessels in mouse models of tumor transplantation [56]. Using an NPC model, Wang et al. found that overexpressed EBV-miR-BART10-5p and hsa-miR-18a upregulated VEGF and HIF-1α in a Spry3-dependent manner and strongly promoted angiogenesis. Moreover, in both in vitro and in vivo NPC models, treatment with iRGD-tagged exosomes enclosing antagomiR-BART10-5p and antagomiR-18a inhibited angiogenesis [135]. Therefore, exosome engineering is a promising tool in RNA-based therapeutics for cancer treatment (Figure 3B). Recently, RNA interference (RNAi)-based strategies, CRISPR/Cas9-mediated circRNA knockout, CRISPR/Cas13-mediated circRNA knockdown and circRNA-induced overexpressed plasmids were developed to target ncRNAs for therapeutic purposes both in vitro and in vivo [136]. In a study, the expression of pro-angiogenic factors in HUVECs was significantly reduced after miR-92a-3p was knocked down in exosomes using an miR-92a-3p inhibitor (miR-92a-3p-i) [137]. Furthermore, because EV-derived ncRNAs perform significant biological functions, specifically targeting EV-derived ncRNAs may be a promising strategy for treating many types of tumors. Currently, many studies aimed at regulating the production of EVs or blocking the uptake of EVs to achieve the goal of treating patients with cancer are underway. Using EVs as a delivery platform is a promising strategy; however, due to high costs and strict ethical regulations, the mass production of EVs is not easy to achieve to develop commercial viability.

Acquired resistance to chemotherapy remains a major obstacle in treating tumors. Previous studies have reported that EV-derived ncRNAs not only impact chemosensitivity but also regulate mechanisms underlying the development and progression of anticancer drug resistance [138,139]. Recent studies have indicated that EV-derived ncRNAs play a pivotal role in anticancer drug resistance. Deng et al. showed that doxorubicin treatment increased MDSC-produced exosomal miR-126a, which rescued doxorubicin-induced MDSC death and promoted tumor angiogenesis in an S100A8/A9-dependent manner [140]. In addition, Fu et al. found that multidrug-resistant HCC Bel/5-FU cells transported exosomal miR-32-5p to Bel7402-sensitive cells and activated the PI3K/AKT pathway, thereby regulating angiogenesis and EMT to further induce multidrug resistance [141]. Additionally, exosomal miR-549a promotes angiogenesis and enhances vascular permeability by regulating the expression of HIF-1α in vascular ECs, thereby promoting tumorigenesis and metastasis in TKI-resistant clear cell renal cell carcinoma [142]. Therefore, overcoming drug resistance during cancer treatment is key to achieving good results. Because some EV-derived ncRNAs can induce drug resistance, EVs may be engineered to be used as ncRNA carriers to overcome chemotherapeutic drug resistance [143]. Studies have shown that EVs can deliver anti-angiogenic ncRNAs, thus providing favorable evidence for their potential use in therapeutic applications. Various ncRNAs, other types of exogenous therapeutic drugs (such as anti-angiogenic drugs), and their combination can be loaded into EVs for cancer treatment. Moreover, EV-derived ncRNAs are considered promising tools for the diagnosis, prognosis, and treatment of various tumors. However, further investigation is required before these therapeutic approaches can be used in clinical settings for cancer treatment.

## 6. Concluding Remarks and Future Perspectives

This review highlights the mechanisms of EV-derived ncRNAs in tumor angiogenesis and anti-angiogenic resistance, their potential as diagnostic biomarkers, and their therapeutic applications. To date, studies have indicated that EV-derived ncRNAs play a versatile role in tumor angiogenesis by remodelling the TME involved in the crosstalk of tumor cells with various immune and stromal cells. Furthermore, these EV-derived ncRNAs can activate several classical signalling pathways by regulating various angiogenesis-related molecules. However, the following questions remain to be addressed to accelerate translational research on EV-derived ncRNAs in tumor diagnosis and treatment: How do EV-derived ncRNAs regulate angiogenesis at the transcriptional and post-transcriptional levels? How do immune cells in the TME, such as NK and B cells, release EVs containing ncRNAs to affect angiogenesis? How can new technologies be used to develop novel therapeutic agents for cancer therapy? More studies should be conducted to investigate EVs derived from different cancer types and to examine their respective roles in tumor angiogenesis. A better understanding of the interactions between tumor and other cells in the TME, such as immune and stromal cells, may provide new insights into EV biology and EV-based therapeutics to improve cancer patient outcomes.

Studies have indicated that EV-derived ncRNAs act as a promising source for diagnostic biomarkers because they are more tissue-specific, and they have more stable characteristics than traditional protein biomarkers. Although significant progress has been made in identifying candidate biomarkers, more large-scale, multi-center clinical trials should be well-designed to verify their utility.

Owing to the unique biological characteristics of EVs, the use of EVs to deliver therapeutic agents is recognized as a new approach with great potential in cancer therapy. Although EV engineering is a prime focus area for research, the findings of most studies are basic. For instance, current study findings on EV engineering are mainly derived from traditional 2-dimensional cell culture systems, which cannot be used to stimulate the TME. Therefore, more reliable models should be developed and validated for investigating anti-angiogenesis. Fortunately, scholars are now creating 3-dimensional organoids made from co-cultures of cancer cells with immune or stromal cells to investigate tumor angiogenesis, which will accelerate the development of new therapeutic approaches for cancer treatment using EVs as natural carriers of drugs and ncRNAs. Therefore, we can expect that more novel RNA-based therapeutic agents will be translated from the bench to the bedside in the future.

## Figures and Tables

**Figure 1 cells-11-00947-f001:**
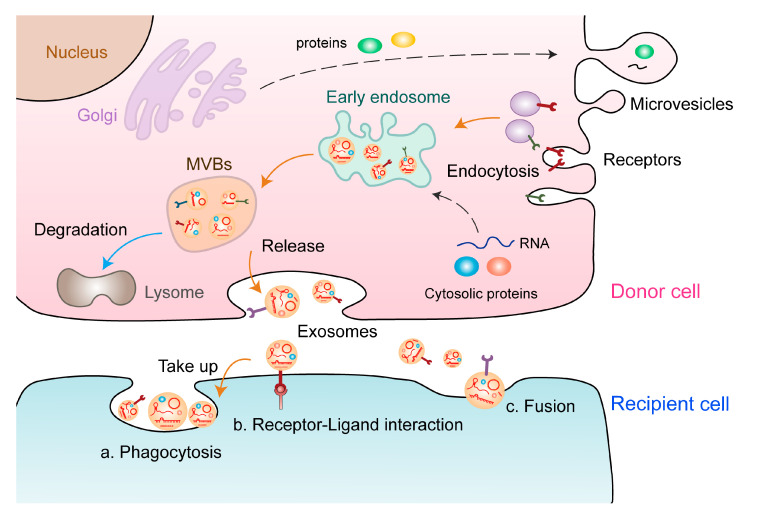
Formation and release of EVs. The formation of EVs involves the following processes: Proteins are transported from the Golgi apparatus or internalised from the cell surface, and nucleic acids are endocytosed and transferred to early endosomes. Early endosomes gradually mature into late endosomes and MVBs, some of which are degraded, whereas others are secreted as exosomes. These exosomes carry multiple biological components, including proteins, lipids, and nucleic acids (e.g., ncRNAs), which are delivered to the recipient cell through different ways: a. phagocytosis, b. receptor–ligand interaction, and c. direct fusion.

**Figure 2 cells-11-00947-f002:**
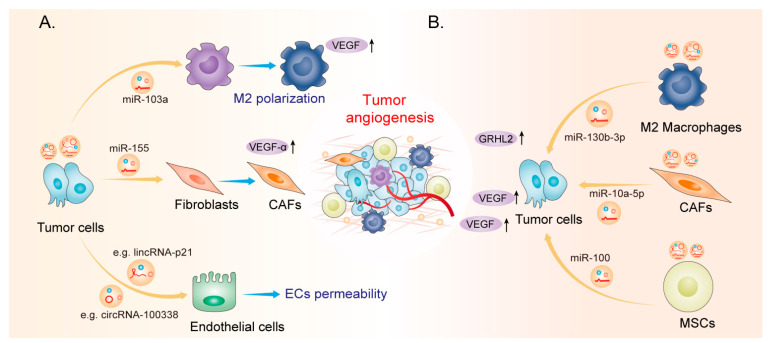
EV-derived ncRNAs regulate cell–cell communication in the angiogenic microenvironment. Tumor cells (**A**), immune cells, and stromal cells (**B**) release EVs containing miRNAs, lncRNAs, circRNAs, and piRNAs, contributing to tumor angiogenesis within the tumor microenvironment. ECs: endothelial cells, CAFs: cancer-associated fibroblasts, and MSCs: mesenchymal stromal cells.

**Figure 3 cells-11-00947-f003:**
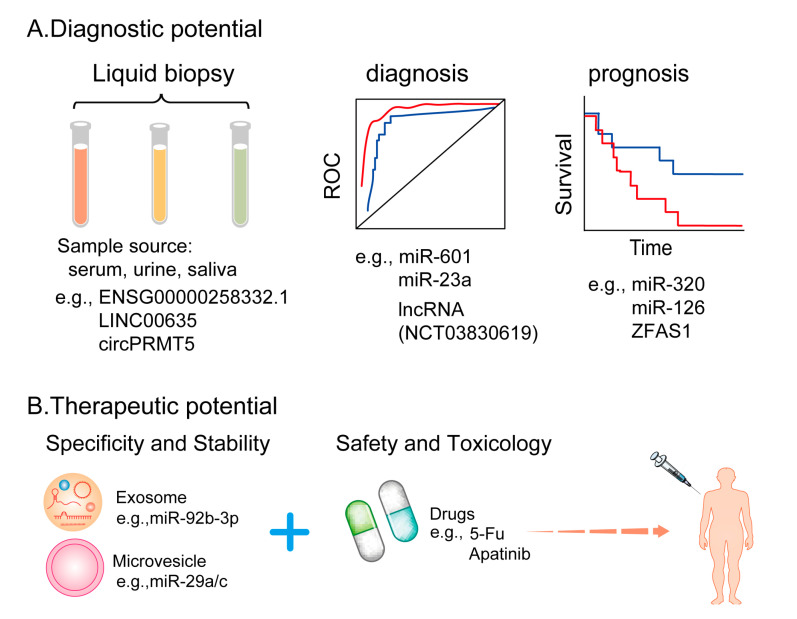
The potential clinical application of EV-derived ncRNAs in tumor angiogenesis (**A**) EV-derived ncRNAs can be detected from patient samples and are potential diagnostic and prognostic biomarkers. (**B**) A combination of targeting EV-derived ncRNAs and using conventional anti-angiogenic agents can enhance therapeutic efficacy.

**Table 1 cells-11-00947-t001:** The emerging roles of EV-derived ncRNAs in tumor angiogenesis.

EV-Derived ncRNAs	Expression	Source Cell	Function and Mechanism	Tumor Type	Reference
miR-155	Upregulated	Tumor cell	Promotes angiogenesis via the c-MYB/VEGF axis	Gastric cancer	[53]
	Upregulated	Tumor cell	Promotes angiogenesis by inhibiting FOXO3a	Gastric cancer	[54]
miR-130a	Upregulated	Tumor cell	Activates angiogenesis by inhibiting c-MYB	Gastric cancer	[55]
miR-135b	Upregulated	Tumor cell	Promotes angiogenesis by inhibiting FOXO1	Gastric cancer	[56]
	Upregulated	Tumor cell	Regulates the HIF/FIH signalling pathway	Multiple myeloma	[57]
miR-23a	Upregulated	Tumor cell	Inhibits PTEN and activates the AKT pathway	Gastric cancer	[58]
	Upregulated	Tumor cell	Increases angiogenesis by inhibiting ZO-1	Lung cancer	[59]
miR-200b-3p	Downregulated	Tumor cell	Enhances endothelial ERG expression	Hepatocellular carcinoma	[60]
miR-25-3p	Upregulated	Tumor cell	Inhibits KLF2 and KLF4, thereby elevating VEGFR2 expression	Colorectal cancer	[61]
miR-1229	Upregulated	Tumor cell	Inhibits HIPK2, thereby activating the VEGF pathway	Colorectal cancer	[62]
miR-183-5p	Upregulated	Tumor cell	Inhibits FOXO1, thereby promoting expression of VEGFA, VEGFAR2, ANG2, PIGF, MMP-2, and MMP-9	Colorectal cancer	[63]
miR-142-3p	Upregulated	Tumor cell	Inhibits TGFβR1	Lung adenocarcinoma	[64]
miR-103a	Upregulated	Tumor cell	Inhibits PTEN, thereby promoting the polarization of M2 macrophages	Lung cancer	[65]
miR-126	Upregulated	MSCs	Upregulates CD34 and CXCR4, thereby promoting expression of VEGF	Lung cancer	[66]
miR-141-3p	Upregulated	Tumor cell	Inhibits SOCS5, thereby activating JAK/STAT3 and NF-κB signalling pathways	Ovarian cancer	[67]
miR-205	Upregulated	Tumor cell	Regulates the PTEN/AKT pathway	Ovarian cancer	[68]
miR-9	Downregulated	Tumor cell	Inhibits MDK, thereby regulating the PDK/AKT signalling pathway	Nasopharyngeal carcinoma	[69]
	Upregulated	Tumor cell	Promotes angiogenesis by targeting COL18A1, THBS2, PTCH1, and PHD3	Glioma	[70]
miR-23a	Upregulated	Tumor cell	Promotes angiogenesis by inhibiting TSGA10	Nasopharyngeal carcinoma	[71]
miR-210	Upregulated	Tumor cell	Enhances tube formation by inhibiting EFNA3	Leukemia	[72]
	Upregulated	Tumor cell	Promotes angiogenesis by inhibiting SMAD4 and STAT6	Hepatocellular carcinoma	[73]
miR-26a	Upregulated	Tumor cell	Inhibits PTEN, thereby activating the PI3K/AKT signalling pathway	Glioma	[74]
miR-27a	Upregulated	Tumor cell	Inhibits BTG2, thereby promoting VEGF, VEGFR, MMP-2, and MMP-9 expression	Pancreatic cancer	[75]
miR-155-5p /miR-221-5p	Upregulated	M2 macrophages	Promotes angiogenesis by targeting E2F2	Pancreatic cancer	[76]
miR-21-5p	Upregulated	Tumor cell	Promotes angiogenesis by targeting TGFBI and COL4A1	Papillary carcinoma	[77]
miR-100	- -	MSCs	Regulates the mTOR/HIF-1α signalling axis	Breast cancer	[78]
miR-21	Upregulated	Tumor cell	Inhibits SPRY1, thereby promoting VEGF expression	Oesophageal squamous cell carcinoma	[79]
	Upregulated	Tumor cell	Inhibits PTEN, thereby activating PDK1/AKT signalling	Hepatocellular carcinoma	[80]
miR-181b-5p	Upregulated	Tumor cell	Inhibits PTEN and PHLPP2, thereby activating AKT signalling	Oesophageal squamous cell carcinoma	[81]
miR-9	Upregulated	Tumor cell	Inhibits S1P, thereby promoting VEGF expression	Medulloblastoma and xenoglioblastoma	[82]
miR-10a-5p	Upregulated	CAFs	Inhibits TBX5, thereby activating Hedgehog signalling	Cervical squamous cell carcinoma	[83]
miR-135b	Upregulated	Tumor cell	Enhances angiogenesis by targeting FIH-1	Multiple myeloma	[57]
miR-130b-3p	Upregulated	M2 macrophages	Regulates the miR-130b-3p/MLL3/GRHL2 signalling cascade	Gastric cancer	[84]
lncGAS5	Downregulated	Tumor cell	Inhibits angiogenesis by regulating the miR-29-3p/PTEN axis	Lung cancer	[85]
lnc-CCAT2	Upregulated	Tumor cell	Promotes VEGFA and TGF-β expression	Glioma	[86]
lnc-POU3F3	Upregulated	Tumor cell	Promotes bFGF, bFGFR, and VEGFA expression	Glioma	[87]
lncRNA RAMP2-AS1	Upregulated	Tumor cell	Promotes angiogenesis through the miR-2355-5p/VEGFR2 axis	Chondrosarcoma	[88]
OIP5-AS1	Upregulated	Tumor cell	Regulates angiogenesis and autophagy through miR-153/ATG5 axis	Osteosarcoma	[89]
FAM225A	Upregulated	Tumor cell	Promotes angiogenesis through the miR-206/NETO2/FOXP1 axis	Oesophageal squamous cell carcinoma	[51]
UCA1	Upregulated	Tumor cell	Promotes angiogenesis through the miR-96-5p/AMOTL2 axis	Pancreatic cancer	[90]
SNHG11	Upregulated	Tumor cell	Promotes angiogenesis through the miR-324-3p/VEGFA axis	Pancreatic cancer	[91]
SNHG1	Upregulated	Tumor cell	Promotes angiogenesis by regulating the miR-216b-5p/JAK2 axis	Breast cancer	[92]
AC073352.1	Upregulated	Tumor cell	Binds and stabilizes the YBX1 protein	Breast cancer	[93]
MALAT1	Upregulated	Tumor cell	Facilitates angiogenesis and predicts poor prognosis	Ovarian cancer	[94]
TUG1	Upregulated	Tumor cell	Promotes angiogenesis by inhibiting caspase-3 activity	Cervical cancer	[95]
LINC00161	Upregulated	Tumor cell	Promotes angiogenesis and metastasis by regulating the miR-590-3p/ROCK axis	Hepatocellular carcinoma	[96]
H19	Upregulated	Cancer stem cell	Promotes VEGF production and release in ECs	Liver cancer	[97]
circSHKBP1	Upregulated	Tumor cell	Enhances VEGF mRNA stability by the miR-582-3p/HUR axis	Gastric cancer	[52]
circRNA-100,338	Upregulated	Tumor cell	Facilitates HCC metastasis by enhancing invasiveness and angiogenesis	Hepatocellular carcinoma	[98]
circCMTM3	Upregulated	Tumor cell	Promotes angiogenesis and HCC tumor growth by the miR-3619-5p/SOX9 axis	Hepatocellular carcinoma	[99]
circ_0007334	Upregulated	Tumor cell	Accelerates CRC tumor growth and angiogenesis by the miR-577/KLF12 axis	Colorectal cancer	[100]
CircFNDC3B	Downregulated	Tumor cell	Inhibits angiogenesis and CRC progression by the miR-937-5p/TIMP3 axis	Colorectal cancer	[101]
circGLIS3	Upregulated	Tumor cell	Induces endothelial cell angiogenesis by promoting Ezrin T567 phosphorylation	Glioma	[102]
piRNA-823	Upregulated	Tumor cell	Promotes VEGF and IL-6 expression	Multiple myeloma	[103]

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
