# Peer review of "Noncoding RNAs of Extracellular Vesicles in Tumor Angiogenesis: From Biological Functions to Clinical Significance"

_cells, 2022, doi:10.3390/cells11060947_

Round 1

Reviewer 1 Report

This review gives a comprehensive overview of the current knowledge regarding the functions of EV-ncRNAs in tumor angiogenesis, and also try to outline their potential clinical applications.

The paper explores the key aspects of the topic in terms of the biology of extracellular vesicles, emerging roles (and regulated signaling pathways) of EV-ncRNAs in tumor angiogenesis and, and the potential clinical applications.

1) In the tumor microenvironment, exosomes mediators of tumor angiogenesis can be released by different cell types (i.e. endothelial cells, stromal cells , and mesenchymal stem cells). In addition, tumor-derived EV-ncRNAs can trigger the proangiogenic potential of circulating angiogenic cells.

Is it possible to classify the cited EV-ncRNAs according to the cell type that release them? Or, at least, outline better this aspect?

2) Some lncRNAs (MALAT1, MANTIS, PUNISHER, MEG3, MIAT, SENCR and GATA6-AS) able to influence angiogenesis are included in Table 1 (and in text).

Indeed, there are reports that "localize" some of them also in extracellular vesicles.

Is there a reason why they are not discussed in the paper?

3) In Table 1, at least 7 ncRNAs  (miR-135b, miR-155, miR-210,miR-23a,miR-27a,miR-9,miRNA-21) are reported two or three times along with with different targets, and sorted by Tumor Type : it should be better to sort by ncRNAs.

Author Response

Response to Reviewer 1 Comments

Comments and Suggestions for Authors

This review gives a comprehensive overview of the current knowledge regarding the functions of EV-ncRNAs in tumor angiogenesis, and also tries to outline their potential clinical applications. The paper explores the key aspects of the topic in terms of the biology of extracellular vesicles, emerging roles (and regulated signaling pathways) of EV-ncRNAs in tumor angiogenesis, and the potential clinical applications.

Point 1:  In the tumor microenvironment, exosomes mediators of tumor angiogenesis can be released by different cell types (i.e. endothelial cells, stromal cells, and mesenchymal stem cells). In addition, tumor-derived EV-ncRNAs can trigger the proangiogenic potential of circulating angiogenic cells. Is it possible to classify the cited EV-ncRNAs according to the cell type that release them? Or, at least, outline better this aspect?

Response: The previous manuscript did not classify clearly the cited EV-derived ncRNAs according to the cell types. In the revised manuscript, we revised and modified the text of section 3, which classifies EV-ncRNAs released from the cell types as follows. 3.1 Tumor cell-derived EVs (Figure 2A); 3.2 immune cells and stromal cell-derived EVs (Figure 2B).  

Point 2: Some lncRNAs (MALAT1, MANTIS, PUNISHER, MEG3, MIAT, SENCR, and GATA6-AS) able to influence angiogenesis are included in Table 1 (and in text). Indeed, there are reports that "localize" some of them also in extracellular vesicles. Is there a reason why they are not discussed in the paper?

Response: In the revised manuscript, we revised and added some EV-derived lncRNAs that influence tumor angiogenesis in Table 1 and content, including MALAT1, H19, SNHG1, MEG3, SNHG11. We have added this sentence as follows on page 9, Lines 244-256.Lines 245-257: Exosomal lncRNAs (such as MALAT1 and RAMP2-AS1) are transferred to recipient HUVECs and eventually promote angiogenesis by increasing VEGF/VEGFR2 expression in epithelial ovarian cancer (OC) and chondrosarcoma, respectively [88,94]. In breast cancer (BC), exosomal SNHG1 promotes the proliferation, migration, and angiogenesis of HUVECs via the  miR-216b-5p/JAK2 axis [92]. In pancreatic cancer (PC), exosomal UCA1 is delivered to HUVECs and promotes angiogenesis via miR-96-5p/AMOTL2 [90]. In GC, exosomal X26nt decreases vascular endothelial cadherin (VE-cadherin) expression by directly binding to the 3'UTR of VE-cadherin mRNA, thereby increasing vascular permeability and angiogenesis in HUVECs [117]. Exosomal AC073352.1 secreted by BC cells is internalised by HUVEC and promotes angiogenesis by interacting with YBX1 and stabilising its protein expression [93]. Exosomal H19 released by cancer stem cells promotes endothelial tube production by upregulating VEGF [97].   

Point 3: In Table 1, at least 7 ncRNAs  (miR-135b, miR-155, miR-210, miR-23a, miR-27a, miR-9, miRNA-21) are reported two or three times along with different targets, and sorted by Tumor Type: it should be better to sort by ncRNAs.

Response: In the revised manuscript, we revised Table 1 according to the ncRNA types and cancer types. We also put together the same ncRNA reported from different publications.

Reviewer 2 Report

In their manuscript Hu and colleagues explore the role of Noncoding RNAs of extracellular vesicles in tumor angiogenesis an highly regulated multi-step process for the formation of new blood vessels that contribute to tumor proliferation that induce malignant transformation of distinct cells by transferring RNAs including noncoding RNA (ncRNAs). They elucidated the functional relevance of extracellular vesicles ncRNAs in tumor angiogenesis summarizing the current research progress on biological functions and underlying their mechanisms of action in several cancer types. They highlighted the potential applications of these ncRNAs as cancer biomarkers and novel therapeutic targets discussing there the possibility to use them to accelerate anti-angiogenic based therapy.

I’ve read with interest the review that nicely covers this interesting topic in a quite comprehensive and clear way. I have just few minor comments and suggestions that would improve this well written piece of work:

- In the paragraph entitled “Characteristic of extracellular vescicles and ncRNA” the authors describe summarily the biological feauters of ncRNAs, despite begin the topic of paper. I would suggest that a deeper illustration would be useful for better understanding of the mechanism of actions, which characterize the different ncRNAs, and the development of resulting therapies.

- Moreover, there are few evidences about role of piRNA in association with extracellular vesicles. Since the manuscript represent a complete review of the topic, it would be would be of interest to the audience a discussion about this class of ncRNA and their potential as therapeutic target in this biological context. To this aim you could include the following references and discuss about them:

Li B, Hong J, Hong M, Wang Y, Yu T, Zang S, Wu Q. piRNA-823 delivered by multiple myeloma-derived extracellular vesicles promoted tumorigenesis through re-educating endothelial cells in the tumor environment. Oncogene. 2019 Jun;38(26):5227-5238. doi: 10.1038/s41388-019-0788-4. Epub 2019 Mar 19. PMID: 30890754.

Peng Q, Chiu PK, Wong CY, Cheng CK, Teoh JY, Ng CF. Identification of piRNA Targets in Urinary Extracellular Vesicles for the Diagnosis of Prostate Cancer. Diagnostics (Basel). 2021 Oct 3;11(10):1828. doi: 10.3390/diagnostics11101828. PMID: 34679526; PMCID: PMC8534571.

- Additionally, I would suggest improving Figure 1 conception meaning adding “protein” under the arrow from Golgi to micro vesicles, defining extracellular elements that are internalized.

- Moreover in Figure 1 legend (line 96) the letter a) b) c) must be included into the Figure.

- Furthermore, in the text, the reference to the Figure 1 (line 85) could be moved to line 72.

- There are few typos along the text that need revisions

Author Response

Response to Reviewer 2 Comments

Comments and Suggestions for Authors

In their manuscript, Hu and colleagues explore the role of Noncoding RNAs of extracellular vesicles in tumor angiogenesis a highly regulated multi-step process for the formation of new blood vessels that contribute to tumor proliferation that induce malignant transformation of distinct cells by transferring RNAs including noncoding RNA (ncRNAs). They elucidated the functional relevance of extracellular vesicles ncRNAs in tumor angiogenesis summarizing the current research progress on biological functions and underlying their mechanisms of action in several cancer types. They highlighted the potential applications of these ncRNAs as cancer biomarkers and novel therapeutic targets discussing there the possibility to use them to accelerate anti-angiogenic based therapy. I’ve read with interest the review that nicely covers this interesting topic in a quite comprehensive and clear way. I have just a few minor comments and suggestions that would improve this well-written piece of work:

  • In the paragraph entitled “Characteristic of extracellular vescicles and ncRNA” the authors describe summarily the biological feauters of ncRNAs, despite begin the topic of paper. I would suggest that a deeper illustration would be useful for better understanding of the mechanism of actions, which characterize the different ncRNAs, and the development of resulting therapies.
  • Point 1:- Moreover, there are few evidences about role of piRNA in association with extracellular vesicles. Since the manuscript represent a complete review of the topic, it would be would be of interest to the audience a discussion about this class of ncRNA and their potential as therapeutic target in this biological context. To this aim you could include the following references and discuss about them: Li B, Hong J, Hong M, Wang Y, Yu T, Zang S, Wu Q. piRNA-823 delivered by multiple myeloma-derived extracellular vesicles promoted tumorigenesis through re-educating endothelial cells in the tumor environment. Oncogene. 2019 Jun;38(26):5227-5238. doi: 10.1038/s41388-019-0788-4. Epub 2019 Mar 19. PMID: 30890754.Peng Q, Chiu PK, Wong CY, Cheng CK, Teoh JY, Ng CF. Identification of piRNA Targets in Urinary Extracellular Vesicles for the Diagnosis of Prostate Cancer. Diagnostics (Basel). 2021 Oct 3;11(10):1828. doi: 10.3390/diagnostics11101828. PMID: 34679526; PMCID: PMC8534571.
  • Response: We thank the reviewer for the expert insights. In the revised manuscript, we added the related reports about the role of EV-derived piRNA in tumor angiogenesis. We have added this sentence as follows on page 3 and page 9 . Lines 117-119: piRNAs, as a type of small RNAs with a length of 21–35 nt, specifically interact with the PIWI protein to perform multifaceted functions in germline development and somatic tissues [29-31].Lines 238-241: In addition, another type of small ncRNAs, piRNA, has also been reported to regulate angiogenesis. piRNA-823 delivered by MM-derived EVs promote angiogenesis by increasing the secretion of VEGF and IL-6 from ECs [104].
  • Point 2:- Additionally, I would suggest improving Figure 1 conception meaning adding “protein” under the arrow from Golgi to microvesicles, defining extracellular elements that are internalized. - Moreover in Figure 1 legend (line 96) the letter a) b) c) must be included into the Figure. Response: In the revised manuscript, we revised Figure 1 according to the reviewer’s good suggestion, including adding “protein” under the arrow from Golgi to microvesicles, adding the letter ‘a) b) c)’.
  • Point 3:- Furthermore, in the text, the reference to the Figure 1 (line 85) could be moved to line 72.
  • Response: As suggested, we have corrected the reference pointed out by the reviewer in the revised manuscript.
  • Point 4:- There are a few typos along with the text that needs revisions
  • Response: As suggested, we have corrected the typos and errors pointed out by the reviewer in the revised manuscript. In order to achieve high standards, the revised manuscript has been polished by editors of BULLET EDITS Language Editing Services.